

# Exact real-time dynamics of single-impurity Anderson model from a single-spin hybridization-expansion

Patryk Kubiczek [1⋆], Alexey N. Rubtsov [2,3] and Alexander I. Lichtenstein [1]

**1** I. Institute for Theoretical Physics, University of Hamburg, Hamburg, Germany
**2** Russian Quantum Center, Skolkovo, Moscow Region, Russia
**3** Department of Physics, Lomonosov Moscow State University, Moscow, Russia

⋆ patryk.kubiczek@physik.uni-hamburg.de

## Abstract

In this work we introduce a modified real-time continuous-time hybridization-expansion quantum Monte Carlo solver for a time-dependent single-orbital Anderson impurity model: CT-1/2-HYB-QMC. In the proposed method the diagrammatic expansion is performed only for one out of the two spin channels, while the resulting effective single-particle problem for the other spin is solved semi-analytically for each expansion diagram. CT-1/2-HYB-QMC alleviates the dynamical sign problem by reducing the order of sampled diagrams and makes it possible to reach twice as long time scales in comparison to the standard CT-HYB method. We illustrate the new solver by calculating an electric current through impurity in paramagnetic and spin-polarized cases.

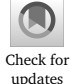
# 1 Introduction

Continuous-time quantum Monte Carlo methods (CT-QMC) are widely used to solve impurity models in equilibrium [1]. Despite their successful application for uncovering short-time nonequilibrium dynamics [2–8], they suffer from an inherent dynamical sign problem preventing an access to longer times. There have been numerous attempts to tame the sign problem in real-time QMC simulations, such as explicit summation over Keldysh indices [9–12], bold-line methods [8, 13–16], or inchworm algorithm [17–21]. Alternatively, a deterministic iterative summation of path integrals is possible [22, 23].

Here we propose a method that alleviates the dynamical sign problem in case of the single-orbital Anderson impurity model (AIM). It is based on the continuous-time hybridization-expansion QMC (CT-HYB-QMC [24]) and can be viewed as a kind of a bold-line QMC. In our method, called CT-1/2-HYB, only the diagrams for one out of two spins need to be sampled (hence the name). The contribution from the other spin can be summed up semi-analytically for each expansion term. The advantage of such an approach, which to our best knowledge has not yet been applied neither to equilibrium nor non-equilibrium calculations, is that the Monte Carlo average expansion order is decreased leading to the reduction of the sign problem. The problem is however that the semi-analytic evaluation of dressed diagrams for a continuous-bath is a non-trivial task. The solution we come up with in this work is to discretize the bath, which limits the spectral resolution and the timescale obtainable within our method.

The paper is organized as follows. In Section 2 we describe CT-1/2-HYB method in detail introducing two possible ways of semi-analytical evaluation of the bold diagrams. In Section 3 we present the strategy for discretizing the bath and results for a current through impurity in a paramagnetic and a spin-polarized case, as well as discuss the computational performance of the algorithm. In section 4 we summarize our findings and draw conclusions.

# 2 Method

## 2.1 Single-spin hybridization-expansion

The system for which CT-1/2-HYB is designed is the single-orbital AIM with general time-dependent parameters. Since the test case we consider in this paper is electric current calculation, it is convenient to introduce a setup with two baths, where each bath (lead) is labelled by index $\alpha$. The Hamiltonian is composed of the local, bath and hybridization parts and reads

$$H(t) = H_{\mathrm{loc}}(t) + \sum_{\sigma} \big[ H_{\mathrm{bath},\sigma}(t) + H_{\mathrm{hyb},\sigma}(t) \big], \tag{1}$$

$$H_{\mathrm{loc}}(t) = \sum_{\sigma} \mathcal{E}_{d\sigma}(t) d_{\sigma}^{\dagger} d_{\sigma} + U(t) d_{\uparrow}^{\dagger} d_{\uparrow} d_{\downarrow}^{\dagger} d_{\downarrow}, \tag{2}$$

$$H_{\mathrm{bath},\sigma}(t) = \sum_{\alpha \in \{-1,1\}} \sum_{p=1}^{N/2} \left( \varepsilon_{p\sigma}(t) + \alpha \frac{\phi(t)}{2} \right) c_{\alpha p\sigma}^{\dagger} c_{\alpha p\sigma}, \tag{3}$$

$$H_{\mathrm{hyb},\sigma}(t) = \sum_{\alpha \in \{-1,1\}} \sum_{p=1}^{N/2} \left( V_{p\sigma}(t) c_{\alpha p\sigma}^{\dagger} d_{\sigma} + \mathrm{h.c.} \right). \tag{4}$$

The method of solution is based on the perturbative expansion of the dynamical partition function $Z$ with respect to hybridization strength on the Keldysh contour $\mathcal{C}$ with an additional imaginary time branch, defined by inverse temperature $\beta$ and maximum evolution time $t_{\mathrm{max}}$ (Fig. 1). The imaginary time branch enables us to start the time-evolution from an equilibrium

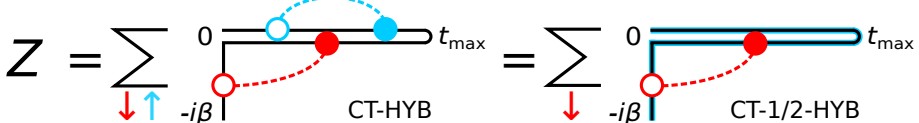

Figure 1: Graphical representation of CT-HYB and CT-1/2-HYB expansions on the Keldysh contour $\mathcal{C}$. The contour starts at $0_+$ at the upper real branch, goes to $t_{\max}$, then goes back to $0_-$ along the lower real branch and eventually reaches $-i\beta$ along the imaginary axis. Full and empty circles denote virtual creation and annihilation processes of impurity electrons. Bold blue line denotes the semi-analytic dressing of spin-down expansion diagrams by all the spin-up processes.

thermal state at $t = 0$. An accessible introduction to this formalism containing imaginary- and real-time evolution can be found e.g. in [25].

Contrary to CT-HYB, the expansion is performed only for one spin species. We will be referring to spin-down as the expansion channel, which is an arbitrary choice made for a convenience of the discussion. In the following we work in the Schrödinger picture: contour-time arguments of operators serve only as contour labels (they do not denote Heisenberg-like evolution). Thus, we expand $Z$ in $H_{\mathrm{hyb}\downarrow}$ obtaining

$$
Z = \mathrm{Tr}\left[\mathcal{T}_{\mathcal{C}}\, e^{-i\int_{\mathcal{C}} dt H(t)}\right] = \mathrm{Tr}\left[\mathcal{T}_{\mathcal{C}}\, e^{-i\int_{\mathcal{C}} dt\left[H(t)-H_{\mathrm{hyb}\downarrow}(t)\right]} e^{-i\int_{\mathcal{C}} dt H_{\mathrm{hyb}\downarrow}(t)}\right]
$$
$$
= \sum_{k=0}^{\infty}(-1)^k \int_{0_+}^{-i\beta} dt_1^{\mathrm{c}}\ldots\int_{t_{k-1}^{\mathrm{c}}}^{-i\beta} dt_k^{\mathrm{c}} \int_{0_+}^{-i\beta} dt_1^{\mathrm{a}}\ldots\int_{t_{k-1}^{\mathrm{a}}}^{-i\beta} dt_k^{\mathrm{a}}
$$
$$
w_{\mathrm{loc+bath}\uparrow}(\{t_m^{\mathrm{c}}\}, \{t_n^{\mathrm{a}}\})\, w_{\mathrm{bath}\downarrow}(\{t_m^{\mathrm{c}}\}, \{t_n^{\mathrm{a}}\}). \tag{5}
$$

The contribution of each diagram, specified by expansion order $k$ along with sets of impurity-electron creation $\{t_m^{\mathrm{c}}\}$ and annihilation times $\{t_m^{\mathrm{a}}\}$, can be represented as a product of the spin-down bath weight

$$
w_{\mathrm{bath}\downarrow}(\{t_m^{\mathrm{c}}\}, \{t_n^{\mathrm{a}}\}) = \mathrm{Tr}_{c_\downarrow}\left[\mathcal{T}_{\mathcal{C}}\, e^{-i\int_{\mathcal{C}} dt H_{\mathrm{bath}\downarrow}(t)} C_\downarrow(t_k^{\mathrm{c}})\ldots C_\downarrow(t_1^{\mathrm{c}}) C_\downarrow^\dagger(t_k^{\mathrm{a}})\ldots C_\downarrow^\dagger(t_1^{\mathrm{a}})\right], \tag{6}
$$

where $C_\downarrow^\dagger(t) \equiv \sum_p V_{p\downarrow}(t) c_{p\downarrow}^\dagger$, and the local weight dressed by spin-up hybridization

$$
w_{\mathrm{loc+bath}\uparrow}(\{t_m^{\mathrm{c}}\}, \{t_n^{\mathrm{a}}\})
$$
$$
= \mathrm{Tr}_{d_\uparrow, d_\downarrow, c_\uparrow}\left[\mathcal{T}_{\mathcal{C}}\, e^{-i\int_{\mathcal{C}} dt\left[H_{\mathrm{loc}}(t)+H_{\mathrm{bath}\uparrow}(t)+H_{\mathrm{hyb}\uparrow}(t)\right]} d_\downarrow^\dagger(t_k^{\mathrm{c}})\ldots d_\downarrow^\dagger(t_1^{\mathrm{c}}) d_\downarrow(t_k^{\mathrm{a}})\ldots d_\downarrow(t_1^{\mathrm{a}})\right]. \tag{7}
$$

The spin-down bath weight can be readily evaluated following the procedure known from CT-HYB method [1]. Defining the hybridization function

$$
\Delta_\sigma(t, t') = \sum_{\alpha p} V_{p\sigma}^*(t) g_{\alpha p\sigma}(t, t') V_{p\sigma}(t'), \tag{8}
$$

where $g_{\alpha p\sigma}(t, t')$ is a free-electron Green function [25]

$$
g_{\alpha p\sigma}(t, t') = \langle \mathcal{T}_{\mathcal{C}}\, c_{\alpha p\sigma}(t) c_{\alpha p\sigma}^\dagger(t')\rangle = -i\left[\theta_{\mathcal{C}}(t, t') - \frac{1}{e^{\beta\epsilon_{p\sigma}(0)} + 1}\right] e^{-i\int_{t'}^{t} d\bar{t}\, \epsilon_{p\sigma}(\bar{t})}, \tag{9}
$$

we obtain

$$
w_{\mathrm{bath}\downarrow}(\{t_m^{\mathrm{c}}\}, \{t_n^{\mathrm{a}}\}) = i^k Z_{\mathrm{bath}\downarrow} \det\left[\Delta_\downarrow(t_m^{\mathrm{c}}, t_n^{\mathrm{a}})\right]_{m,n=1,\ldots,k}, \tag{10}
$$

where $Z_{\mathrm{bath}\downarrow} = \mathrm{Tr}_{c_\downarrow}\left[\mathcal{T}_{\mathcal{C}}\, e^{-i\int_{\mathcal{C}} dt H_{\mathrm{bath}\downarrow}(t)}\right]$ is a diagram-independent constant. Namely, for each diagram one needs to evaluate a determinant of a $k \times k$ spin-down hybridization matrix.

## 2.2 Dressed local weight from discretized bath

The novel aspect of CT-1/2-HYB is the presence of the spin-up hybridization dressing in the local weight, Eq. (7), which incorporates both local and spin-up bath degrees of freedom. This auxiliary system is governed by the Hamiltonian $H_0 = H_{\text{loc}} + H_{\text{bath}\uparrow} + H_{\text{hyb}\uparrow}$ and is effectively non-interacting since $H_0$ conserves $n_\downarrow = \langle d_\downarrow^\dagger d_\downarrow \rangle$. The contour-time dependence of $n_\downarrow$ is specified by the the locations of operators $d_\downarrow^\dagger$: $\{t_m^c\}$ and $d_\downarrow$: $\{t_n^a\}$. Because $n_\downarrow$ can equal only 0 or 1, the only non-vanishing contributions to the dressed local trace come from the following orderings of the spin-down operators:

- $-i\beta > t_k^c > t_k^a > t_{k-1}^c > \ldots > t_2^a > t_1^c > t_1^a > 0_+$ (permutation sign: 1),

- $-i\beta > t_k^a > t_k^c > t_{k-1}^a > \ldots > t_2^c > t_1^a > t_1^c > 0_+$ (permutation sign: $(-1)^k$).

In the following we assume that only those two orderings of contour-times $\{t_m^c\}, \{t_n^a\}$ are allowed. Note that for $k = 0$ both contributions have to be summed, corresponding to constant $n_\downarrow = 0$ or 1 on the whole contour.

The dressed local weight can be represented as

$$w_{\text{loc+bath}\uparrow}(\{t_m^c\}, \{t_n^a\}) = \varphi(\{t_m^c\}, \{t_n^a\}) \cdot \det\left[ \mathbb{1} + \mathcal{U}(-i\beta, 0_+; \{t_m^c\}, \{t_n^a\}) \right]. \tag{11}$$

The factor $\varphi$ takes into account the permutation sign and the phases resulting from tracing over impurity spin-down occupations. If we denote the effective contour-time-dependent spin-down impurity occupation by $n_\downarrow(t) \equiv n_\downarrow(t; \{t_m^c\}, \{t_n^a\})$ then

$$\varphi(\{t_m^c\}, \{t_n^a\}) = \begin{cases} 1 & \text{if } t_1^c > t_1^a \\ (-1)^k & \text{else} \end{cases} \cdot e^{-i \int_\mathcal{C} dt \, \mathcal{E}_{d\downarrow}(t) n_\downarrow(t)}. \tag{12}$$

The determinant part represents the trace of the evolution operator of the spin-up impurity and the spin-up bath subject to an effective contour-time-dependent single-particle Hamiltonian $h_\uparrow(t)$

$$h_\uparrow(t) = \begin{bmatrix} \mathcal{E}_{d\uparrow}(t) + U(t) n_\downarrow(t) & V_{1\uparrow}^*(t) & \cdots & V_{N\uparrow}^*(t) \\ V_{1\uparrow}(t) & \varepsilon_{1\uparrow}(t) & \cdots & 0 \\ \vdots & \vdots & \ddots & 0 \\ V_{N\uparrow}(t) & 0 & 0 & \varepsilon_{N\uparrow}(t) \end{bmatrix}. \tag{13}$$

The single particle evolution operator $\mathcal{U}$ is given by

$$\mathcal{U}(t, t'; \{t_m^c\}, \{t_n^a\}) = \mathcal{T}_\mathcal{C} \, e^{-i \int_t^{t'} d\bar{t} \, h_\uparrow(\bar{t})} \tag{14}$$

and the relation between the trace of the many-body evolution operator and the single-particle evolution operator is provided by the identity valid for arbitrary fermions $f$ and a non-interacting Hamiltonian $h(t)$

$$\text{Tr}_f \left[ \mathcal{T}_\mathcal{C} \, e^{-i \int_\mathcal{C} dt \sum_{ab} f_a^\dagger h_{ab}(t) f_b} \right] = \det\left[ \mathbb{1} + \mathcal{T}_\mathcal{C} \, e^{-i \int_\mathcal{C} dt \, h(t)} \right]. \tag{15}$$

Note that this identity is also used in the determinant QMC [26], which is similarly based on the summation of determinants of matrices in the single particle basis. It is known that care must be taken while evaluating determinants of the form of Eq. (15) for large $\beta$ due to exponentially large and small matrix elements. We followed the advice from [27] to stabilize the necessary matrix multiplications and evaluation of determinants.

Because the contour-time dependence of $n_\uparrow(t)$ is step-wise constant we can take advantage of a convenient segment picture [1] to obtain a more practical representation of $\phi$ and $\mathcal{U}$. We

call a *segment* a time interval during which $n_\downarrow = 1$, i.e. after a creation time and before an annihilation time. Then we have

$$\varphi(\{t_m^c\}, \{t_n^a\}) = \begin{cases} 1 & \text{if } t_1^c > t_1^a \\ (-1)^k & \text{else} \end{cases} \cdot \prod_{\text{segments}} e^{-i \int_{\text{seg.}} dt\, \mathcal{E}_{d\downarrow}(t)} \tag{16}$$

and

$$\mathcal{U}(-i\beta, 0_+; \{t_m^c\}, \{t_n^a\}) = \begin{cases} u_1(-i\beta, t_k^c) u_0(t_k^c, t_k^a) u_1(t_k^a, t_{k-1}^c) \ldots u_0(t_1^c, t_1^a) u_1(t_1^a, 0_+) & \text{if } t_1^c > t_1^a \\ u_0(-i\beta, t_k^a) u_1(t_k^c, t_k^a) u_0(t_k^c, t_{k-1}^a) \ldots u_1(t_1^a, t_1^c) u_0(t_1^c, 0_+) & \text{else,} \end{cases} \tag{17}$$

where $u_{n_\downarrow}$ is a single-particle time-evolution matrix

$$u_{n_\downarrow}(t, t') = \mathcal{T}_C \exp\left( -i \int_{t'}^t d\bar{t} \begin{bmatrix} \mathcal{E}_{d\uparrow}(\bar{t}) + U(\bar{t}) n_\downarrow & V_{1\uparrow}^*(\bar{t}) & \cdots & V_{N\uparrow}^*(\bar{t}) \\ V_{1\uparrow}(\bar{t}) & \varepsilon_{1\uparrow}(\bar{t}) & \cdots & 0 \\ \vdots & \vdots & \ddots & 0 \\ V_{N\uparrow}(\bar{t}) & 0 & 0 & \varepsilon_{N\uparrow}(\bar{t}). \end{bmatrix} \right).$$

## 2.3 Discretizing contour-time instead of bath

Another approach to evaluate the dressed local weight is to discretize the contour-time instead of the bath. Within the path integral-formalism one can integrate out the spin-up bath from Eq. (7) and work directly with the hybridization function $\Delta_\uparrow$ defined by Eq. (8). After calculating traces over the bath and spin-down impurity degrees of freedom we obtain

$$w_{\text{loc+bath}\uparrow}(\{t_m^c\}, \{t_n^a\}) = \varphi(\{t_m^c\}, \{t_n^a\}) \cdot Z_{\text{bath}\uparrow} \cdot Z_{d\uparrow}(\{t_m^c\}, \{t_n^a\}), \tag{18}$$

where $\varphi$ is the same as in Eq. (12) and (16), $Z_{\text{bath}\uparrow}$ is a constant and $Z_{d\uparrow}$ is given by the path integral

$$Z_{d\uparrow}(\{t_m^c\}, \{t_n^a\}) = \int \mathcal{D}\left(d_\uparrow, d_\uparrow^*\right) e^{i \int_C \int_C dt dt'\, d_\uparrow^*(t) \mathcal{G}_\uparrow^{-1}(t, t') d_\uparrow(t')}, \tag{19}$$

with

$$\mathcal{G}_\uparrow^{-1}(t, t') = \mathcal{G}_{0\uparrow}^{-1}(t, t') - \Delta_\uparrow(t, t'), \tag{20}$$

$$\mathcal{G}_{0\uparrow}^{-1}(t, t') = \left( i\frac{\partial}{\partial t} - \mathcal{E}_{d\uparrow}(t) - U(t) n_\downarrow(t) \right) \delta_C(t, t'), \tag{21}$$

and where $n_\downarrow(t)$ is determined by the sets of contour-times $\{t_m^c\}, \{t_n^a\}$. When $\mathcal{G}_\uparrow$ is understood as a matrix in time domain, the path integral $Z_{d\uparrow}$ is given by the formula [28]

$$Z_{d\uparrow}(\{t_m^c\}, \{t_n^a\}) = \det\left[ i\mathcal{G}_\uparrow^{-1} \right]. \tag{22}$$

Because it is undesirable to discretize the time derivative operator we can recast this formula using Eq. (20)

$$Z_{d\uparrow}(\{t_m^c\}, \{t_n^a\}) = z_{d\uparrow}(\{t_m^c\}, \{t_n^a\}) \cdot \det\left[ \mathbb{1} - \mathcal{G}_{0\uparrow} \circ \Delta_\uparrow \right], \tag{23}$$

where $\circ$ denotes contour-time convolution and $z_{d\uparrow}$ is a generalized partition function for a system with time-local action which can be explicitly evaluated

$$z_{d\uparrow}(\{t_m^c\}, \{t_n^a\}) = \det\left[ i\mathcal{G}_{0\uparrow}^{-1} \right] = 1 + e^{-i \int_C dt\, [\mathcal{E}_{d\uparrow}(t) + U(t) n_\downarrow(t)]}. \tag{24}$$

For each diagram a generalized Green function $\mathcal{G}_{0\uparrow}$ should be calculated as a solution to Eq. (21). However, this has to be done for several contour-time spacings $\Delta t$ as an extrapolation of

the result down to $\Delta t = 0$ is required. Provided the convolution in Eq. (23) is done carefully as described in [29] the error of the discrete approximation should scale as $\Delta t^2$.

We have decided not to follow this method due to its high computational cost. Nevertheless, this approach might be of a future interest since it allows one for an introduction of a retarded interaction $U_{\uparrow\downarrow}(t, t')$ by adding an additional term $-U_{\uparrow\downarrow}(t, t')n_{\downarrow}(t)$ to the left hand side of Eq. (20), which is only a small numerical effort.

## 2.4  Measurement of observables

The partition function $Z$ from Eq. (5) can be viewed as a sum of the weights $w(C)$ of diagrams $C = \{k; \{t_m^c\}, \{t_n^a\}\}$ for all expansion orders $k$ and contour-times $\{t_m^c\}, \{t_n^a\}$

$$Z = \sum_C w(C). \tag{25}$$

The space of diagrams $C$ is sampled within a Markov-chain Monte Carlo simulation. We have implemented the segment removal and addition moves, which guarantee the ergodicity of the sampling algorithm, and the shift moves (shifting the beginning or the end of a segment), which decrease the autocorrelation time. A detailed description of those moves along with Metropolis-Hastings acceptance rates can be found in [1]. We use fast updates for determinants of the hybridization matrices (bath weight) but calculate the dressed local weight explicitly for each Monte Carlo diagram.

The diagram weights discussed above can be in general any complex numbers. To work around this issue, we sample the diagrams with probabilities given by the absolute value of weights while simultaneously measuring their average (complex) sign. The expectation value of an observable $A$ is given by

$$\langle A \rangle \equiv \frac{1}{Z}\mathrm{Tr}\Big[\mathcal{T}_C\, e^{-i\int_C dt H(t)}A\Big] = \frac{\sum_C A(C)w(C)}{\sum_C w(C)} = \frac{\langle A\,\mathrm{sgn}\rangle_{\mathrm{MC}}}{\langle \mathrm{sgn}\rangle_{\mathrm{MC}}}, \tag{26}$$

where $A(C)$ is the expectation value of $A$ in diagram $C$, the complex sign is defined as

$$\mathrm{sgn}(C) = \frac{w(C)}{|w(C)|} \tag{27}$$

and the Monte Carlo average follows the prescription

$$\langle A \rangle_{\mathrm{MC}} = \frac{\sum_C A(C)|w(C)|}{\sum_C |w(C)|}. \tag{28}$$

It is known that CT-HYB-QMC for a single-impurity Anderson model does not suffer from the fermionic sign problem, i.e. the average sign on the imaginary axis is always one. However, the presence of real-time branches causes the dynamical sign problem. For each diagram having operators on the real branch there exists another one which has exactly the same weight but the opposite sign. This results from the fact that operator closest to $t_{\mathrm{max}}$ on the real axis can be moved freely from the upper to the lower branch (or conversely) and this operation changes only the sign of the diagram. This means that the average complex sign $\langle \mathrm{sgn} \rangle$ is a real quantity which equals the probability of having no operators on real-time branches. Practically, the average sign decreases exponentially with $t_{\mathrm{max}}$ which effectively prohibits an accurate evaluation of observables using Eq. (26) for large $t_{\mathrm{max}}$. To overcome the inaccuracy of the results we have to scale the total number of Monte Carlo samples exponentially with $t_{\mathrm{max}}$ until it is feasible.

In the following we present explicit measurement formulas for certain observables of interest.

The time-dependent impurity spin-down occupation is the average of the diagram-dependent function $n_\downarrow(t; \{t_m^c\}, \{t_n^a\})$ where $t$ can lie either on the upper or the lower real-time branch

$$n_\downarrow(t) = \left\langle n_\downarrow(t; \{t_m^c\}, \{t_n^a\}) \right\rangle. \tag{29}$$

The measurement of spin-up observables requires the evaluation of the elements of occupation matrix for the system of coupled spin-up impurity and bath. Namely, it can be shown

$$n_\uparrow(t) = \left\langle 1 - [\mathbb{1} + \mathcal{U}(t, 0_+)\mathcal{U}(-i\beta, t)]_{00}^{-1} \right\rangle. \tag{30}$$

This result follows from the general formula analogous to the one found in [26]

$$\frac{\mathrm{Tr}_f \left[ \mathcal{T}_C \, e^{-i \int_C dt \sum_{ab} f_a^\dagger h_{ab}(t) f_b} f_i(t) f_j^\dagger(t') \right]}{\mathrm{Tr}_f \left[ \mathcal{T}_C \, e^{-i \int_C dt \sum_{ab} f_a^\dagger h_{ab}(t) f_b} \right]} = \tag{31}$$

$$(-1)^{\theta_C(t',t)} \left[ \left( \mathbb{1} + \mathcal{T}_C \, e^{-i \int_{0_+}^t d\bar t \, h(\bar t)} \mathcal{T}_C \, e^{-i \int_t^{-i\beta} d\bar t \, h(\bar t)} \right)^{-1} \mathcal{T}_C \, e^{-i \int_{t'}^t d\bar t \, h(\bar t)} \right]_{ij}. \tag{32}$$

The spin $\sigma$ electron current from lead $\alpha$ to the impurity is given by

$$I_\sigma^\alpha(t) = -2 \, \mathrm{Im} \sum_p^{N/2} V_{p\sigma}(t) \left\langle c_{\alpha p\sigma}^\dagger d_\sigma \right\rangle(t), \tag{33}$$

which can be obtained by solving the Heisenberg equation of motion for the impurity particle number operator $d_\sigma^\dagger d_\sigma$ and employing the continuity equation

$$\frac{dn_\sigma}{dt} = \sum_\alpha I_\sigma^\alpha(t). \tag{34}$$

While evaluation of the expectation value of the current operator requires a reference solution within the standard CT-HYB algorithm (see e.g. Eq. (60) in [5]), it is possible to extract the full spin-up single-particle occupation matrix directly in CT-1/2-HYB and by that all the relevant matrix elements for the spin-up current. Similarly to (30)

$$\left\langle c_{\alpha p\uparrow}^\dagger d_\uparrow \right\rangle(t) = -\left\langle [\mathbb{1} + \mathcal{U}(t, 0_+)\mathcal{U}(-i\beta, t)]_{0, \alpha p}^{-1} \right\rangle, \tag{35}$$

where $\alpha p$ is the index of the column corresponding to $p$th electronic level from bath $\alpha$. In practice, one measures the whole spin-up single-particle occupation matrix as defined in (30) and (35), from which the observables of interest can be constructed.

## 3 Results

### 3.1 Discrete representation of the bath

Before presenting the results, we show in detail the parameters of the bath we have chosen for our simulation. We want to model a bath defined by the following time- and spin-independent coupling function

$$\Gamma_\alpha(\omega) = \frac{\Gamma}{\left(1 + e^{\nu(\omega - D)}\right)\left(1 + e^{-\nu(\omega + D)}\right)}, \tag{36}$$

which represents a flat band of half-width $D = 4$ with a soft cutoff controlled by parameter $\nu = 3$, coupled to the impurity with the coupling strength $\Gamma = 1$. However, within the discrete

bath approach, only a peak-structured coupling function can be constructed

$$\Gamma_\alpha(\omega) = \pi \sum_p^{N/2} |V_p|^2 \delta(\omega - \varepsilon_p). \tag{37}$$

In order to approximate the continuous bath-impurity coupling function, Eq. (36), by a discrete one, Eq. (37), we apply the *equal weight method* from Ref. [30], obtaining equal hybridization parameters $V_p$

$$V_p = \sqrt{\frac{2}{\pi N} \int d\omega \, \Gamma_\alpha(\omega)} = \sqrt{\frac{2}{\pi N} \cdot \Gamma D (1 + \coth(\nu D))}. \tag{38}$$

The resulting discrete representations of the bath for several values of bath size $N$ are presented in Fig. 2.

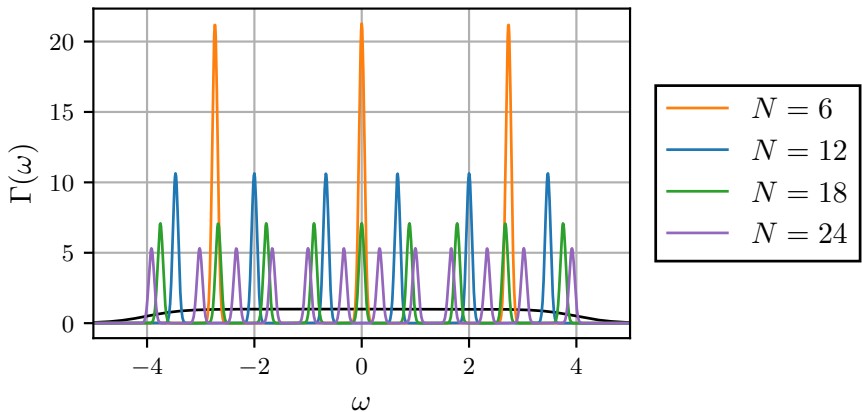

Figure 2: Discrete approximation to the right/left impurity-bath coupling functions $\Gamma_\alpha(\omega) = \Gamma(\omega)$ for different (total) numbers of bath sites $N$ (a Gaussian broadening of width 0.1 was used for the purpose of the picture). The solid black line is the continuous coupling function from Eq. (36)

## 3.2 Current through impurity

We measure the electric current in a simulation of a voltage quench from $\phi(t = 0) = 0$ to $\phi(t > 0) = 2$. In order to analyze the convergence of result with increasing bath size, we perform QMC calculations for several numbers $N$ of bath sites. We set inverse temperature $\beta = 5$ and Coulomb interaction on the impurity $U = 4$. We consider the half-filled case, such that the (time-independent) local energy level

$$\mathcal{E}_{d\sigma} = -\frac{U}{2}. \tag{39}$$

In Fig. 3 we plot the spin-up electric current through impurity as a function of time. Due to the particle-hole symmetry the left hand side of Eq. (34) vanishes, thus currents from the left and right are equal up to the sign. Also, in the paramagnetic case the total current can be recovered by multiplication by factor of 2. For reference we also provide a result of an exact-diagonalization calculation for $N = 6$ which confirms the validity of our method.

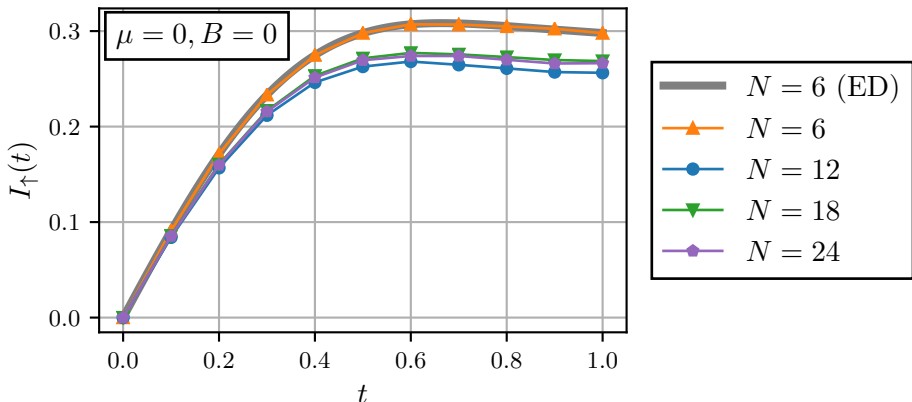

Figure 3: Spin-up current in the half-filled paramagnetic case for various bath sizes $N$

As also observed in a similar study [18], the current relatively quickly (around 1 unit of $\Gamma^{-1}$) reaches its steady state value. We note a quick convergence of the result with $N$. The difference between results for $N = 18$ and $N = 24$ is only minimal.

It is instructive to look at the comparison of the average expansion order and the average Monte Carlo sign between CT-1/2-HYB and CT-HYB, which we present in Fig. 4 for $N = 12$ (the dependence of those quantities on $N$ is negligible). In the paramagnetic case CT-1/2-HYB leads to the reduction of the average expansion order by a factor of two since only spin-down diagrams have to be sampled while spin-down and spin-up diagrams are contributing to $Z$ with equal weights. Noteworthy, the expansion order grows linearly with time and the rate of growth is smaller by a factor of 2. According to the established exponential decrease of the average sign with the growing expansion order [4,5] twice as long time scales can be reached within CT-1/2-HYB method as compared to CT-HYB, when the smallest average sign feasible for obtaining the result is fixed.

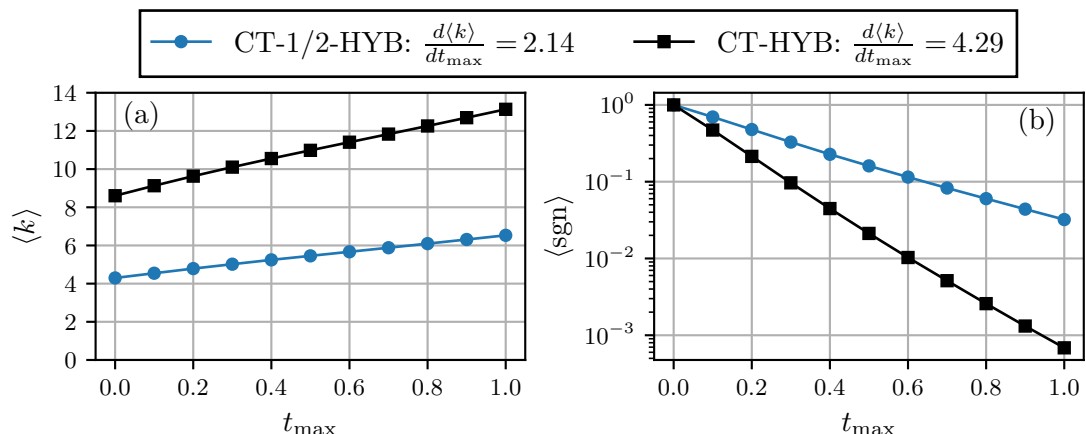

Figure 4: (a) Average expansion order as a function of $t_{max}$ for bath size $N = 12$. The growth rate for CT-1/2-HYB is twice as low as for CT-HYB (estimated values based on a linear fit starting from $t = 0.4$). (b) Average Monte Carlo sign as a function of $t_{max}$

## 3.3 Effect of magnetic field

A valid research question is whether the CT-1/2-HYB method can lead to an even better performance improvement over CT-HYB in presence of a magnetic field breaking the SU(2) symmetry of the model. In such a case, the average expansion orders in the two spin channels can differ, and hence one can choose to expand $Z$ in the channel with the lower expansion order. In the following, we modify the definition of the local energies $\mathcal{E}_{d\sigma}$ such that it takes into account non-zero chemical potential $\mu$ and magnetic field $B$

$$\mathcal{E}_{d\sigma} = -\frac{U}{2} - \mu - \frac{\sigma}{2}B. \tag{40}$$

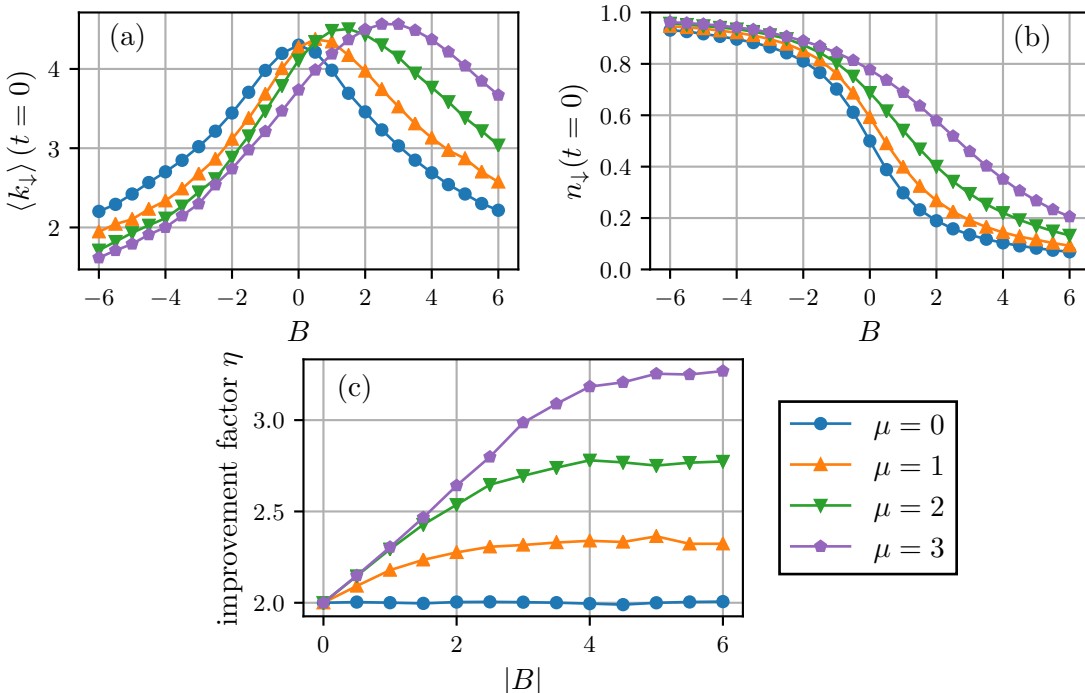

Figure 5: (a) Equilibrium average CT-1/2-HYB expansion order vs. magnetic field $B$. (b) Equilibrium spin-down impurity level filling $n_\uparrow$ as a function of magnetic field $B$ and chemical potential $\mu$. Spin-up filling is obtained by the mirror symmetry around $B = 0$. (c) Improvement factor $\eta$ as a function of $|B|$. All the results are for $N = 12$.

Fig. 5(a) shows the average expansion orders for the imaginary-time (equilibrium) CT-1/2-HYB as a function of $B$ for several values of $\mu$. The maximum spin-down average order occurs when the spin-down impurity occupation is close to 0.5, which is evident from Fig. 5(b). For further analysis, it helps to realize that the following relation holds

$$\langle k_\uparrow \rangle (B) = \langle k_\downarrow \rangle (-B), \tag{41}$$

where $\langle k_\uparrow \rangle$ is a hypothetical average expansion order in case the hybridization expansion was performed in the spin-up channel. Then, we define the improvement factor $\eta$ as

$$\eta^{-1} = \frac{\min\left(\langle k_\downarrow \rangle, \langle k_\uparrow \rangle\right)}{\langle k_\downarrow \rangle + \langle k_\uparrow \rangle}, \tag{42}$$

where the sum $\langle k_\downarrow \rangle + \langle k_\uparrow \rangle$ is a hypothetical average expansion order in the standard CT-HYB algorithm. We plot the equilibrium improvement factor as a function of $|B|$ in Fig. 5(c). In the

paramagnetic case the improvement factor $\eta$ is always 2. For $\mu = 0$ the improvement factor also equals 2 and does not depend on $B$, which results from the particle-hole symmetry. Only by simultaneous introduction of non-zero $\mu$ and $B$ the average expansion orders can be made different in the two spin channels, leading to $\eta > 2$.

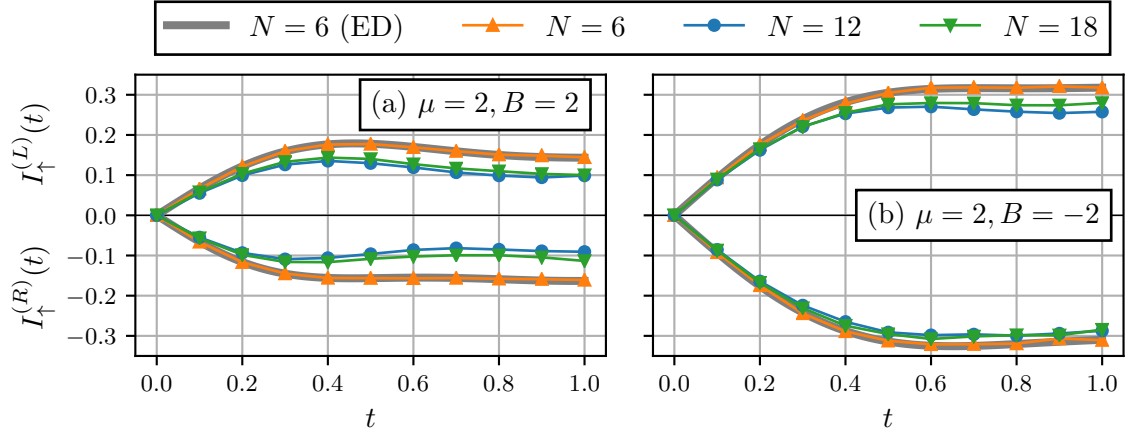

Figure 6: Spin-up currents to the left and right bath for $\mu = 2$ and (a) $B = 2$ or (b) $B = -2$. Fig. (a) can be understood as spin-down currents for $B = -2$, just as Fig. (b) can be viewed as spin-down currents for $B = 2$

While Fig. 5(c) proves that CT-1/2-HYB can more than halve the average expansion order on the imaginary axis, it is not yet clear if the same applies to the real-time evolution. Ultimately, it is the growth of the expansion order with the evolution time that is causing the dynamical sign problem. That is why we decided to estimate this quantity for the calculation of time-dependent current in the presence of chemical potential and magnetic field. We choose $\mu = 2$ and $B = 2, -2$, which is a case where an improvement factor in equilibrium of around 2.5 can be achieved.

In the polarized case currents from the right and from the left bath are different. Fig. 6 presents the left and the right spin-up currents in the presence of magnetic fields $B = -2, 2$. The spin-up current is lower for $B = 2$ since then the spin-up impurity level lies almost at the edge of the bath which hinders transport dynamics.

Fig. 7 shows the growth of the expansion order with time for the paramagnetic case compared with the two polarized cases. Contrary to the imaginary-time calculation, the expansion order grows at almost the same rate irrespective of chemical potential and magnetic field. Noteworthy, for $B = -2$ the growth rate of the average expansion order is slightly higher than for other cases, even though the initial average order (for $t = 0$) is the lowest in this case.

The different behaviors of the average expansion orders in imaginary and real-time expansions were noted previously in [5]. The average expansion order on the Keldysh contour has no interpretation as a quantity proportional to the kinetic energy and is thus only weakly influenced by the position of the impurity level relative to the conduction bath.

## 3.4 Computational performance

The biggest computational burden of the method comes from the need to overcome the dynamical sign problem. For that reason we have scaled the number of Monte Carlo steps exponentially with $t_{max}$ in order to keep the error approximately constant for all times. We have been able to generate enough samples such that the statistical errors are smaller than size of markers on all the plots presented in this work.

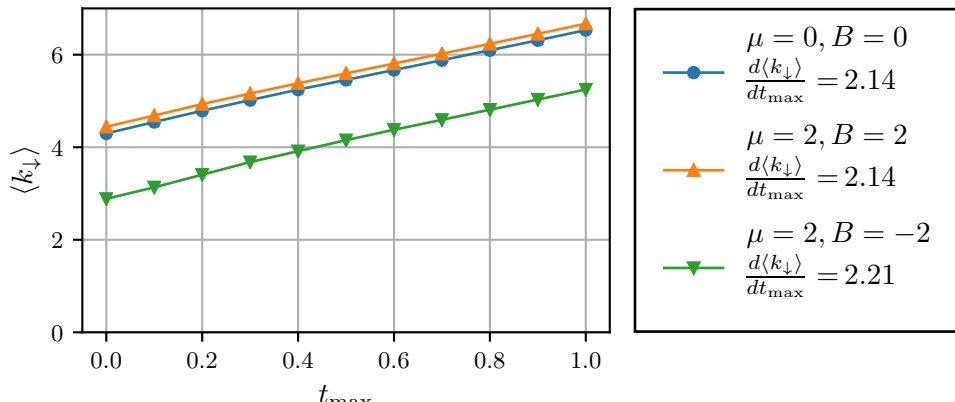

Figure 7: Average CT-1/2-HYB expansion order vs. $t_{\max}$ for three sets of parameters $\mu$, $B$. Although the initial (equilibrium) average expansion order depends on the magnetic field $B$ and chemical potential $\mu$, the growth rate of the expansion order is hardly influenced

The dependence of the average sign on the physical parameters is the same as in the standard CT-HYB method on a Keldysh contour. Namely, keeping the hybridization strength as energy unit, the sign is largely unaffected by interaction strength, chemical potential or magnetic field but decreases with bath bandwidth [4].

Now we discuss the computational complexity of diagram evaluations. For the discretized bath approach the number of bath sites $N$ plays the most important role. Generally, the expected scaling of this approach is $\mathcal{O}(\langle k_\uparrow \rangle N^3)$ and results from multiplications single-particle evolution matrices of size $(N+1) \times (N+1)$, whose average number scales linearly with $\langle k_\uparrow \rangle$. The actual scaling can deviate from this prediction as certain implementation-dependent optimizations might be possible.

For the discretized time approach the number of contour time steps $N_t$ determines the computational scaling. Both the solution of the Dyson equation [25] and the evaluation of the determinant in time-space will scale as $\mathcal{O}(N_t^3)$.

Moreover, it takes on average $\langle k_\uparrow \rangle$ Monte Carlo moves to decorrelate subsequent measurements. This implies an additional scaling factor of $\langle k_\uparrow \rangle$ [13]. The average order $\langle k_\downarrow \rangle$ is linear in inverse temperature $\beta$ and $t_{\max}$: $\langle k_\downarrow \rangle = c_\beta \beta + c_t\, t_{\max}$. Leaving the dynamical sign problem aside and neglecting the cost of bath weight evaluation, the overall theoretical complexity of the discretized bath approach is $\mathcal{O}(\langle k_\downarrow \rangle^2 N^3)$ and that of the discretized contour-time approach is $\mathcal{O}(\langle k_\downarrow \rangle N_t^3)$ (where $N_t$ is the number of time-steps, including those on the imaginary branch).

The computational scaling of our implementation of diagram evaluations can be learned from Fig. 8. The benchmark problem is the evaluation of the paramagnetic current from section 3.2, where discretized bath approach has been utilized.

Fig. 8(a) compares CPU time per Monte Carlo (MC) step in the CT-HYB algorithm against the CT-1/2-HYB results for different numbers $N$ of bath sites. The cost of a single MC step in CT-HYB algorithm is lower for all the considered values of $N$, pointing to a huge computational effort related to the evaluations of the dressed local weight. The approximately linear growth of the CPU time per step with $t_{\max}$ is due to an increasing average order $\langle k_\downarrow \rangle$.

It is also constructive to understand the scaling of the necessary computational resources with growing $N$. In Fig. 8(b) we present the CPU time per MC step as a function of $N$ for a fixed $t = 1$. The fitted computational complexity is $\mathcal{O}(aN^2 + bN^3)$ with $a \gg b$.

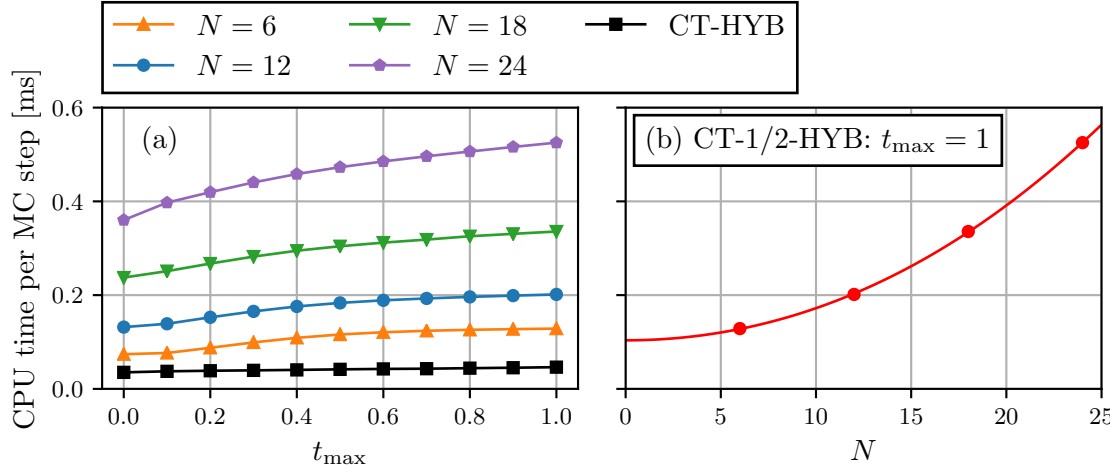

Figure 8: (a) CPU time per Monte Carlo step for CT-HYB and CT-1/2-HYB for several values of $N$ as a function of $t_{\max}$. (b) CPU time per step for CT-1/2-HYB as a function of $N$ for $t_{\max} = 1$ and a fit of the form $f(N) = c + aN^2 + bN^3$

The most general version of our implementation of CT-1/2-HYB involves a precomputation of all $u_0(t, t')$ and $u_1(t, t')$ on a two-dimensional time-grid and subsequent interpolations during a Monte Carlo run. Unfortunately, this approach can quickly lead to very high memory demands, especially for large $N$. However, when the time-dependence of Hamiltonian parameters is step-wise, which is the case for the voltage quench, the time-evolution generated by the operators $u_{n_\downarrow}$ can be performed in their eigenbases. For this purpose, one needs to store only a small set of eigenenergies and rotation matrices, which not only solves the problem of high memory requirement but also avoids the interpolation-related numerical inaccuracy and presumably improves performance. We have applied this strategy for the calculations presented here, which possibly explains the $\mathcal{O}(N^2)$ contribution to the scaling due to presence of diagonal matrices.

## 4   Conclusion

CT-1/2-HYB-QMC allows one to solve the single-orbital Anderson impurity model exactly on twice as longer time scales than standard CT-HYB-QMC due to a reduction of the average expansion order. The growth rate of the average expansion order with $t_{\max}$ is independent of the location of impurity spin-up and spin-down levels in the considered time-evolution after the voltage quench, which rules out an even better performance gain in cases with non-zero chemical potential and magnetic field.

It is possible to extend this approach to multi-orbital impurity models with density-density interactions. If all the orbitals are interacting, the improvement factor will diminish with the number of fermionic flavors (combined spin and orbital numbers) $n$ as $\frac{n}{n-1}$. This results from the fact that the semi-analytical evaluation of the diagrams will be possible for only one out of $n$ flavors. Moreover, in this case the method can only be applied to Hamiltonians conserving orbital and spin quantum numbers, which poses a serious limitation to the modelling of realistic materials. However, if the interaction matrix is sparse, also some single-particle mixing terms can be allowed between non-interacting flavors, while the improvement factor can be better than $\frac{n}{n-1}$ as more flavors can be treated semi-analytically.

We note that the cost of the CT-HYB inchworm algorithm [17] increases exponentially with number of orbitals due to an exponential increase in the size of many-body Hilbert space, which

does not depend on the sparsity of interactions. On the other hand, in CT-1/2-HYB the increase in computational cost with the number of orbitals is in principle only polynomial. The main challenge of a real-time QMC is however the dynamical sign problem, which still leads to an exponential CPU-time scaling with $t_{\max}$ in CT-1/2-HYB and only to a polynomial scaling in case of inchworm algorithm. Even though CT-1/2-HYB-QMC may significantly extend accessible timescales for multi-orbital models with sparse interactions, it is still subject for future analysis to understand whether it will be enough to study any phenomena of interest.

It is worthwhile to consider whether the CT-1/2-HYB could be combined with bold or inchworm QMC. Their strength is that dressed many-body propagators can be used to evaluate Monte Carlo diagrams. Those propagators are actually of no use in CT-1/2-HYB as an attempt to include many-body corrections to CT-1/2-HYB would undermine the basic assumption about the effective single-particle dynamics. Nevertheless, the inchworm expansion could be done around propagators dressed by spin-up processes in the spirit of CT-1/2-HYB. In contrast to pure CT-1/2-HYB, some spin-up diagrams would be missed in such an inchworm procedure and would still have to be sampled separately. It is unclear at the moment how the efficiency of the inchworm algorithm would be affected by use of dressed propagators since no such attempts have been reported.

An interesting single-orbital application of CT-1/2-HYB method as an impurity solver for DMFT could be investigation of the cross-over behavior between Falicov-Kimball and Hubbard models, by tuning the spin-down hopping from zero to the value of the spin-up hopping. Our method is especially suitable for studying this problem since the zeroth order term of 1/2-HYB expansion corresponds to Falicov-Kimball-like impurity.

## Acknowledgements

We acknowledge the use of Armadillo library [31] for linear-algebra operations in the core of CT-1/2-HYB code and of QuSpin package for ED benchmark [32]. The calculations were performed on the HLRN-4 supercomputer of the North-German Supercomputing Alliance.

**Funding information**  The work has been supported by SFB 925 "Light induced dynamics and control of correlated quantum systems" financed by DFG (P.K., A.I.L.) and by a joint DFG/RNF grant no. 16-42-01057/LI 1413/9-1 (A.N.R., A.I.L.).

## A  Code

Our open-source implementation of the CT-1/2-HYB-QMC impurity solver is available at https://github.com/patryk-kubiczek/noneq-cthalfhyb.

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
