# Peer review of "Exact real-time dynamics of single-impurity Anderson model from a single-spin hybridization-expansion"

_SciPost Physics, doi:SciPost Phys. 7, 016 (2019)_

## Round 1 · Referee Report · Anonymous (Referee 1) · 2019-5-10

Strengths

1—Novel and interesting methodological idea to improve convergence in an important class of computational physics problems.

2—Shown to work in practice, and the resulting improvement is well understood.

3—Possibly compatible with other recent ideas in the field.

4—Expected to be competitive with existing methods near the crossover to Falicov–Kimball.

Weaknesses

1—The method was not tested in a regime where it should be advantageous. This is a minor weakness, and it is acceptable to leave this for a later paper focusing on the physics.

2—The frequency resolution of the lead density of states is sacrificed compared to competing methods.

Report

The manuscript reports on a novel modification of the real time continuous time Monte Carlo algorithm in the hybridization expansion (CT-HYB) for impurity models. The main idea, implemented here for a single impurity Anderson model, is to expand in the hybridization of only one spin species rather than both. One then obtains dressed but effectively noninteracting (Falicov-Kimball) dynamics for the other spin, the action of which can be efficiently evaluated. This is reminiscent of the bold-line Monte Carlo in references [11] and [12] in both spirit and performance, but is clearly distinct.

Two ways of evaluating the effective local action over the remaining spin are proposed. One of these requires discretization of the bath degrees of freedom and is similar in form to what appears in auxiliary field Monte Carlo methods. This was found to be more efficient here. The other requires a time discretization and is more expensive computationally, but has the advantage that it can be extended to cases with retarded local interactions. This could potentially be very important for e.g. GW+DMFT.

In practice, the average expansion order is reduced by a factor of 2, effectively doubling the reachable timescale when compared to standard CT-HYB. This factor is shown to be largely unaffected by a combination of a magnetic field and chemical potential, but is expected to decrease for multiorbital impurity models.

The method is interesting, novel and has some promising immediate applications (the crossover between the Falicov–Kimball and Hubbard models is mentioned). The manuscript is clear and the theory and results are well presented. I therefore recommend publication after some minor modifications listed below.

Requested changes

1—What is the theoretical computational scaling of the method with the relevant physical and numerical parameters, in both approaches to obtaining the dressed local weight? Is the $O(N^{3})$ result in figure 8 understood?

2—Where the summation over Keldysh indices is mentioned in the introduction, please see also arxiv:1904.11969 and arXiv:1903.11646.

3—Where bold-line CT-HYB is mentioned in the introduction, please see also 10.1103/PhysRevB.82.075109, 10.1103/PhysRevB.84.085134 and 10.1103/physrevlett.116.036801 (currently reference 8).

4—More work that should probably be mentioned in the same context as points 2 and 3 is the literature on iterative summation of path integrals by D. Segal and M. Thorwart.

5—Since the method presented here is essentially orthogonal to e.g. the bold, inchworm and (possibly) Keldsyh summation ideas, it may be combined with these other advances. If so, that would be a major point in favor of the manuscript. Could the authors briefly comment on whether this is thought to be feasible?

---

## Round 2 · Referee Report · Anonymous (Referee 1) · 2019-6-24

Report

The revised manuscript has been substantially improved, especially by the more complete discussion of computational complexity and by the interesting discussion regarding connections to other methods. I recommend publication with no further changes.

---

## Round 2 · Referee Report · Anonymous (Referee 2) · 2019-7-23

Strengths

1) New idea/computational approach for dynamics of quantum impurity models, relevant topics in condensed matter physics/mesoscopic physics/material science etc. 2)Longer time scales (by a factor 2) can be reached as compared to standard Hybridization Expansion CTQMC 3)Method is clearly explained in the theory part and in the application, including the limitations, the computational complexity, benchmarks etc.. 4)The basic idea (expanding around one single spin flavour) could lead to interesting developments both conceptual and practical, related to Falikov Kimball physics..

Weaknesses

1)The need for discretizing the bath to evaluate the diagrams for the up spin can be rather severe, expecially to study transport and nonequilibrium effects. It is not completely clear whether the gain in longer time scales accessible is not cancelled off by the need of finite size baths, particularly in interesting and challenging (Kondo) regimes.
2)The gain of a factor 2 with respect to Hyb-CTQMC is good, at the same time recent developments in QMC has resulted in new algorithms that surpass the standard approach (Inchworm, resummation techniques,..).
This is however a minor weakness since in principle the idea of this new approach could be potentially combined with other methods, as the author comment toward the end..

Report

In this work, the authors present a new QMC approach to solve the real-time dynamics of quantum impurity models, a relevant subject of research in condensed matter physics several with potential applications.

The basic idea is to reduce the number of diagrams which are stochastically sampled (and which lead to the infamous sign problem) by evaluating (resumming) certain diagrams semi-analytically. Specifically, the authors consider a spinful Anderson Model and expand the Keldysh partition function only in the down spin. The result is that each diagram is slightly less trivial to evaluate but there are factor 2 less diagrams to sample on average.

The main outstanding challenge that this paper raises in my opinion is how to evaluate efficiently and accurately those diagrams, which correspond to Falikov-Kimball like impurity problems.
The authors propose two methods (discretization of bath size or discretization of time domain) and test only one of them, obtaining good results.

It would be interesting to explore more the second approach: I suspect that using the segment picture and the fact that for QMC sampling only the ratio of weights in Eq.18 between adjacent configuration is needed, one could further simplify the algebra. In addition, Falikov Kimball impurity problems allow an analytical solution (at least for certain non equilibrium problems) and it could be beneficial to explore this direction as well.

To conclude, in my opinion the paper presents an interesting and promising method for a tough and relevant condensed matter problem. The paper is clear and well written, explaining advantages and limitations of the current implementation and could therefore lead to further developments in the community. I am in favor of publication on SciPost without further changes.
  • validity: top
  • significance: high
  • originality: top
  • clarity: top
  • formatting: perfect
  • grammar: excellent

Author:  Patryk Kubiczek  on 2019-07-24  [id 573]

(in reply to Report 2 on 2019-07-23)

We thank the reviewer for their positive feedback on our work. We believe the report accurately identified how the community could benefit from the new method (combination with other approaches, further developments using discretization of time domain), as well as the method's weaknesses (possibly no access to Kondo physics, relatively small improvement factor in comparison to other methods).

---

## Round 2 · Author Response

We thank the reviewer for their positive evaluation of our article and for the valuable comments, which we hereafter address.

1 - We have added a detailed discussion of computational complexity into section 3.4. It reads O(<k>^2 N^3) for discretized bath and O(<k> N_t^3) for discretized time approach.

2 - We have added the proposed references. arXiv:1904.11969 appeared shortly before the paper has been sent and we have not had a chance to look into it. arXiv:1903.11646 reports on extension of the Keldysh summation method to previously unattainable parameter regions by conformal transformations of expansion variable. Since this development has far-reaching consequences for the applicability of the Keldysh summation method, we agree it should be cited.

3 - We have included the recommended references. Even though 10.1103/PhysRevB.82.075109 does not deal with real-time dynamics it sets the basis of the bold QMC method. 10.1103/PhysRevB.84.085134 provides bold QMC calculations of impurity current which is of great relevance to our work. We also agree that 10.1103/physrevlett.116.036801 should be mentioned as an application of the bold-line method.

4 - Although iterative path-integral summation methods are not Monte Carlo methods per se, we do agree they share many common features since the motivation behind them was to overcome the sign problem through a deterministic summation of diagrams. As you suggest, we have mentioned iterative summation of path-integrals as a method alternative to QMC in the introduction, citing the following papers: 10.1002/pssb.201349187 and 10.1103/PhysRevB.82.205323.

5 - We support the idea of discussing compatibility of CT-1/2-HYB with bold and inchworm methods. In principle, combining CT-1/2-HYB with bold or inchworm method is non-trivial, since the latter use many-body propagators and the former uses single-particle propagators. Nevertheless, inchworm algorithm around propagators dressed by spin-up processes in the spirit of CT-1/2-HYB is possible. However, in contrast to pure CT-1/2-HYB some spin-up diagrams will still be missed in the inchworm procedure and will have to be sampled separately. We have added a relevant discussion to the conclusion part of the paper.

---

## Round 2 · List of Changes

1 - mentioned iterative summation of path integrals (section 1)
2 - added references: arXiv:1904.11969, arXiv:1903.11646, 10.1103/PhysRevB.82.075109, 10.1103/PhysRevB.84.085134, 10.1002/pssb.201349187, 10.1103/PhysRevB.82.205323 (section 1)
3 - corrected typo: −U↑↓(t,t′)n↓(t)n↓(t') was changed to −U↑↓(t,t′)n↓(t) (section 2.3)
4 - extended and reorganized the discussion of the computational complexity (section 3.4)
5 - added a discussion of compatibility of CT-1/2-HYB with inchworm and bold algorithms (section 4)

---

## Editorial Decision

published